# Construction and Application of LSTM-Based Prediction Model for Tunnel Surrounding Rock Deformation

Yongchao He [1] and Qiunan Chen [2,*]

1    School of Resource & Environment and Safety Engineering, Hunan University of Science and Technology, Xiangtan 411201, China
2    Hunan Province Key Laboratory of Geotechnical Engineering for Stability Control and Health Monitoring, Hunan University of Science and Technology, Xiangtan 411201, China
*    Correspondence: nmmwm1@163.com

**Abstract:** Tunnel surrounding rock deformation is a significant issue in tunnel construction and maintenance and has garnered attention from both domestic and international scholars. Traditional methods of predicting tunnel surrounding rock deformation involve fitting monitoring and measuring data, which is a laborious and resource-intensive process with low accuracy when predicting data with significant fluctuations. A deep learning approach can improve monitoring efficiency and accuracy while reducing labor costs. In this study, taking an actual tunnel project as an example, a long short-term memory (LSTM) network model was constructed based on the recurrent neural network algorithm with deep learning to model and analyze the tunnel monitoring and measurement data, and the model was used to analyze and predict the vault settlement of the tunnel. LSTM is a type of artificial neural network architecture that is commonly used in deep learning applications for sequence prediction tasks, such as natural language processing, speech recognition, and time-series forecasting. In predicting data with smaller fluctuations, the maximum error is 4.76 mm, the minimum error is 0.03 mm, the root mean square error is 2.64, and the coefficient of determination is 0.98. In predicting data with larger fluctuations, the maximum error is 8.32 mm, the minimum error is 0.13 mm, the root mean square error is 4.42, and the coefficient of determination is 0.88. The average error of the LSTM network model is 2.16 mm. With the growth of the prediction period, the prediction results become more and more stable and closer to the actual vault settlement, which provides a reliable reference for introducing the LSTM prediction method with deep learning to tunnel construction and promoting tunnel construction safety.

**Keywords:** tunnel engineering; deformation prediction; deep learning; long short-term memory (LSTM)

## 1. Introduction

In recent times, China has formulated and advanced its "new infrastructure" strategy, which includes the development of 5G networks, big data centers, artificial intelligence, and the industrial Internet, all of which are considered novel forms of infrastructure. China is the world's largest tunnel construction country [1] and more and more tunnels pass through its complex terrain. In the process of highway tunnel construction, challenges include the poor regional geological conditions of the tunnel, improper choice of blasting parameters, and changes in surrounding rock conditions. To overcome these challenges, artificial intelligence (AI) technologies have been increasingly utilized in tunnel engineering construction, specifically in the area of deep learning for tunnel stability assessment and disaster risk evaluation [2,3]. In order to ensure the safety of tunnel excavation, it is necessary to monitor the surrounding rock of the tunnel and accurately predict the change trend of the tunnel. In 1984, H.H. Instein of the Massachusetts Institute of Technology put forward a paper on the application of artificial intelligence in rock mechanics [4]. In 1985, Fairhurst proposed to use

fuzzy mathematics combined with an expert system to solve the tunnel support problem [5]. In 2006, G.E. Hamilton published deep learning research on reducing data dimensions [6]. In China, Zhang et al. [7] took the lead in introducing artificial intelligence theory into rock mechanics and geotechnical engineering and predicted the mechanical properties of rocks. Song et al. [8] proposed the observation and research method of underground rock strata dynamics, the prediction and control mechanical model, and roof prediction and control technology. Feng et al. [9] put forward the concept of intelligent rock mechanics, found out the internal relations contained in engineering instance data, used these relations to make a reasonable judgment on tunnel stability, and established an intelligent system for rockburst risk estimation. Chen et al. [10] pointed out that modern information technology such as big data should be used to strengthen the construction of the tunnel database and establish intelligent data monitoring. Chen et al. [11] summarized the main research progress of stability analysis, slope intelligent monitoring, and slope intelligent prediction of artificial intelligence technology in geotechnical engineering, and put forward prospects for solving slope stability problems with artificial intelligence technology. Wang et al. [12] carried out the physical and mechanical test research of soft rock, and combined with finite element numerical simulation, analyzed and predicted the deformation of tunnel surrounding rock. Jiang [13] used artificial intelligence technology to predict rock mechanical behavior and proposed the particle swarm optimization support vector machine model for rock mechanical behavior prediction.

Experts have conducted a lot of research on tunnel deformation prediction methods, including theoretical research [14–19], such as constitutive models, plasticity, and damage and viscoplasticity theory. Huang et al. [20] derived a plastic damage scheme for the expansion of undrained spherical cavities in rock media. Plastic damage was considered and the Cam-Clay (MCC) model was modified. Numerical simulation research [21–24] includes DLSM, DEM, FEM, and the peri-dynamic model. Ma et al. [25] conducted a numerical study of the effect of karst caves on tunnel stability using the apparent lattice spring model (DLSM). Huang et al. [26] adopted a stabilization method including surface grouting and internal tunnel grouting. The mechanism of interval lining damage and the effect of stabilization were studied by numerical simulation and analysis of field monitoring data. In terms of machine learning prediction research [27–33], Hu et al. [34], based on the LSTM network, built the Seq2Seq model and proposed the application method of pretreatment of measured settlement data. Early prediction models for tunnel deformation were primarily based on the theory of continuum mechanics. However, the uncertain nature of boundary conditions has led some scholars to employ numerical simulation predictions using finite element software. Nevertheless, the anisotropic nature of rock, faults, and fragmentation can result in deviations between predicted and actual tunnel deformation and groundwater conditions. In light of these challenges, deep learning techniques have been developed to predict tunnel surrounding rock deformation. However, prior studies that utilized multiple tunnel data samples obtained from the literature lack specificity for individual tunnels. To address this issue, this study focuses on a particular tunnel construction project and collects monitoring data during construction. Using LSTM networks, a targeted prediction model is constructed and validated through comparisons with existing models. This model is then applied to the tunnel, providing a new method and technical reference for monitoring tunnel surrounding rock deformation.

The available research results have made a significant contribution to our understanding of tunnel deformation prediction methods. In this paper, we construct LSTM network models based on recurrent neural network algorithms in deep learning to study their effectiveness in predicting the deformation of the Yuanbaoshan Tunnel of Yunnan Li-Xiang Railway. By combining the existing literature and the research of this group, the prediction and analysis of the vault settlement in two different sections of this tunnel are carried out. This study shows that the LSTM prediction model is able to predict the vault settlement during the tunnel construction, and by comparisons with traditional prediction curves, we find that the accuracy and stability of the prediction are more consistent with the actual

measured data in the field, both for data with smaller and larger fluctuations. Predicting tunnel deformation problems with the LSTM network model can promote the practical application of artificial intelligence in tunnel engineering, as well as reduce labor costs and improve monitoring accuracy. It provides a new method and new technology reference for monitoring tunnel deformation.

## 2. Materials and Methods

### 2.1. Recurrent Neural Network

Deep learning is a methodology that involves the transformation of original data features through multiple steps to acquire a feature representation, which is then input into the prediction function to obtain the final output [35]. Figure 1 illustrates that deep learning models primarily comprise neural network models, including recurrent neural networks that possess short-term memory capabilities. In a recurrent neural network, neurons can receive both the information of other neurons as well as their own information, forming a network structure with a loop. The network structure typically consists of an input layer, a hidden layer, and an output layer.

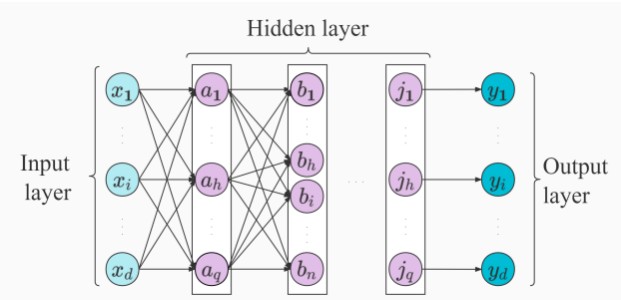

**Figure 1.** Deep learning structure.

Suppose that given an input value, $x_t = (x_1, x_2, \cdots x_t)$, the recurrent neural network can extract features through the hidden layer $h_t$. $h_t$ is calculated as follows:

$$h_t = f(s_0, s_t) \tag{1}$$

where $h_0 = 0$, $f(\cdot)$ is a nonlinear function. As shown in Figure 2, the output layer biases the output results according to different hidden layer weights and itself.

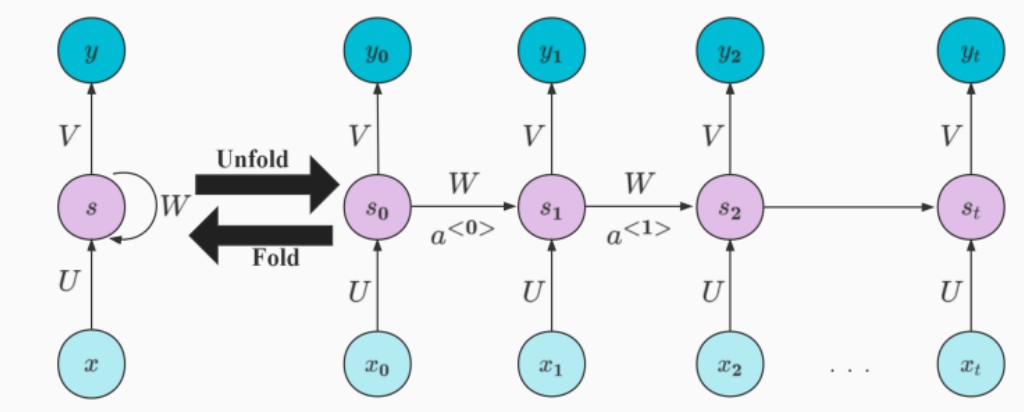

**Figure 2.** Recurrent neural network structure.

### 2.2. LSTM Prediction Model

Due to the problems of vanishing gradients and exploding gradients in traditional recurrent neural networks, it is difficult to learn the data parameters of remote nodes.

Therefore, this study adopts its improved long short-term memory (LSTM) model. As shown in Figure 3, LSTM has a memory function, which can associate the information on time series, find out the features, and carry out long-term learning. It was proposed by Hochreiter and Schmidhuber [36].

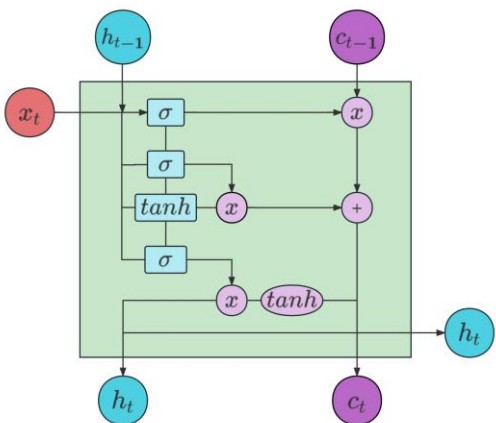

**Figure 3.** LSTM structure.

All recurrent neural networks have the form of a chain of repeating modules. In a traditional recurrent neural network, the structure of repeating modules is very simple, while in LSTM, the structure of repeating modules is very complex, with four neural network layers, and they interact in a special way. The calculation of a single neuron in the LSTM repeat module includes two parts: neural network state update and output value calculation. There are three gate functions in the neuron, which are the input gate, the forgetting gate, and the output gate. The input value, memory value, and output value are controlled by the gate function.

The forgetting gate controls the amount of information discarded by the neural network state at the current moment. The calculation process of forgetting gate is as follows:

$$f_t = \sigma\left(W_f \cdot [h_{t-1}, x_t] + b_f\right) \tag{2}$$

where $f_t$ is the output of the forgetting gate, $h_{t-1}$ is the hidden state at the last moment, and the proportion of information forgotten is controlled by the information fusion $W_f$, $b_f$, and sigmoid function of $\sigma$.

The input gate is divided into two parts, which are the input value of the input gate it and the new input:

$$i_t = \sigma(W_i \cdot [h_{t-1}, x_t] + b_i) \tag{3}$$

$$\widetilde{C}_t = \tanh(W_c \cdot [h_{t-1}, x_t] + b_c) \tag{4}$$

where the output value of the tanh function is between $-1$ and 1, the input gate filters input layer information, and the calculation process of the input gate is as follows:

$$C_t = f_t \cdot C_{t-1} + i_t \cdot \widetilde{C}_t \tag{5}$$

where $C_t$ is the neural network status updated at the current time. The calculation process of the output gate and hidden state is as follows:

$$o_t = \sigma(W_o \cdot [h_{t-1}, x_t] + b_o) \tag{6}$$

$$h_t = o_t \cdot \tanh(C_t) \tag{7}$$

### 2.3. Tunnel Surrounding Rock Deformation Prediction Model

Tunnel surrounding rock deformation prediction is a complex nonlinear problem. Considering that tunnel surrounding rock deformation data are time-series data, the LSTM model is more suitable for tunnel surrounding rock deformation prediction with high timeliness requirements than other models. As shown in Figure 4, combined with the monitoring data of the tunnel surrounding the rock deformation vault settlement, the parameters of the prediction model were analyzed, and we proposed the tunnel surrounding rock deformation prediction model of the LSTM network. Where $x_0, x_1 \cdots x_t$ is the input vault settlement data, $y_0, y_1 \cdots y_t$ is the predicted result of the LSTM model, $s_0, s_1 \cdots s_t$ is the $t$ LSTM neuron of the hiding layer, $C_{t-1}$ is the state of the hiding layer at the previous time, and $h_{t-1}$ is the state of the hiding layer at the previous time.

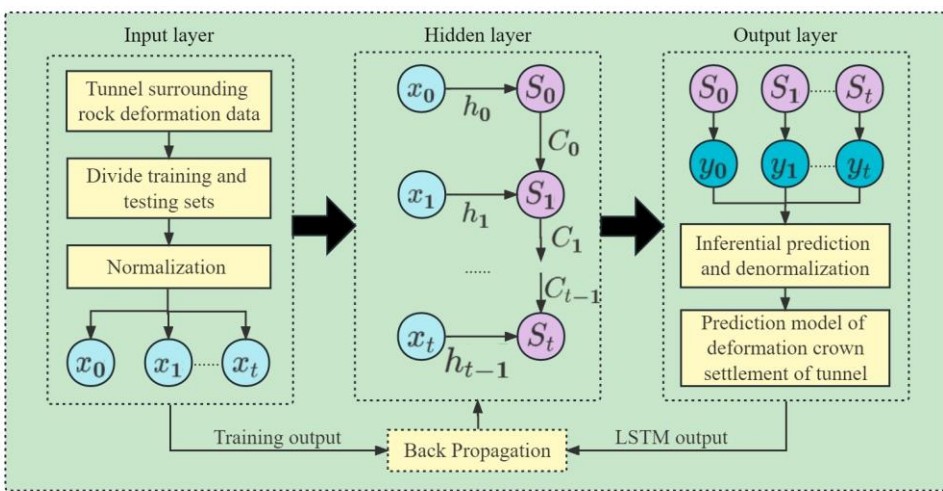

**Figure 4.** LSTM tunnel surrounding rock deformation prediction model.

## 3. Model Establishment and Parameter Selection

### 3.1. Model Establishment and Optimization

The pytorch framework has the advantage of being open source, with simple modules, flexible code, and multiple resources and development. The tunnel surrounding rock deformation prediction model of the LSTM network is selected to run under the pytorch3.6 framework with the CPU version built under the Windows operating system. The Adam optimization algorithm is used in network training. The Adam algorithm can optimize the learning rate of the loss function. Based on 110 sets of engineering data, the tunnel surrounding rock deformation prediction model of the LSTM network is used to train and test the data. A total of 78 sets of data were used for training and 32 sets were used for testing. The learning rate in the training process of the model is set to 0.001, and the total number of sample data used in each training is 21, including 20 training samples and 1 label sample. For example, take the vault settlement value from 1 August to 19 August, $(x_0, x_1, \cdots x_{19})$ as 20 training samples, and the arch settlement $x_{20}$ on 20 August is the label sample. After the training samples were imported into the LSTM model for forward reasoning, the current output value was calculated, and the MSE was calculated for the output value and $x_{20}$. According to MSE, the weight of neurons in LSTM was adjusted by a backpropagation algorithm. The above process is repeated constantly, and the model with the lowest loss moment is taken as the final LSTM prediction model.

### 3.2. Data Sets and Evaluation Indicators

Before building the database of the training and testing network, the arch settlement of the tunnel surrounding rock deformation monitoring data is preprocessed. Pretreatment can speed up the training of neural network models and prevent the gradient explosion in the training process. The surrounding rock deformation monitoring data during tunnel

construction are normalized so that their value domain is distributed within [0, 1]. The Min-MaxScaler function in Python's machine learning library scikit-learn is used for normalized preprocessing of input data; the calculation process is as follows:

$$\widetilde{x} = \frac{x - \min(x)}{\max(x) - \min(x)} \tag{8}$$

During the normalization process, the maximum and minimum values of the samples in the training set should be recorded. After the completion of the model training, the output value of the model should be de-normalized to restore the output to the measured predicted value.

In order to find out suitable model calculation parameters, this paper constructs a tunnel surrounding rock deformation prediction database containing 110 groups of tunnel engineering instances based on the pre-treated surrounding rock deformation monitoring data during tunnel construction. The settlement position of the vault is on the wall of the tunnel after excavation, and monitoring measurement is completed after each excavation and before the next cycle according to the code requirements.

To evaluate the accuracy of the LSTM network model in predicting the arch settlement, the root mean square error ($RMSE$) and determination coefficient ($R^2$) were used to evaluate the accuracy of the prediction model.

$$\text{RMSE} = \sqrt{\frac{1}{m} \sum_{i=1}^{m} (h(x_i) - y_i)^2} \tag{9}$$

$$R^2 = 1 - \frac{\sum\limits_{i=1}^{m} (y_i - h(x_i))^2}{\sum\limits_{i=1}^{m} (y_i - \overline{y})^2} \tag{10}$$

## 4. Engineering Application and Prediction Results Analysis

### 4.1. Engineering Overview

To determine suitable model calculation parameters, the Yuanbaoshan Tunnel of the Yunnan Li-Xiang Railway was selected as the case study. The tunnel's depth of burial is mostly over 300 m, and it passes through a mainly layered carbonitic SLATE stratum. The strata strike is primarily parallel to the tunnel's direction, and the dip angle ranges from 75° to 90°. Due to regional structural factors, the rock mass is susceptible to deformation and exhibits significant joint development, moderate weathering, and fissure water. As a result of the deformation of the tunnel surrounding rock, the supporting structure of the tunnel also undergoes significant deformation. Table 1 presents the relevant tunnel parameters.

**Table 1.** Tunnel parameters.

| The Surrounding Rock Grade | Tunnel Length (km) | Excavation Section Area (m²) | Maximum Vault Subsidence (mm) | Maximum Buried Depth (m) | Strata Inclination (°) |
|---|---|---|---|---|---|
| V | 10.6 | 60.8 | 190 | 687 | 75~90 |

To verify the prediction effect of the established model algorithm on the settlement data of the tunnel vault, typical sections (a) and (b) were selected for testing, as shown in Figures 5 and 6.

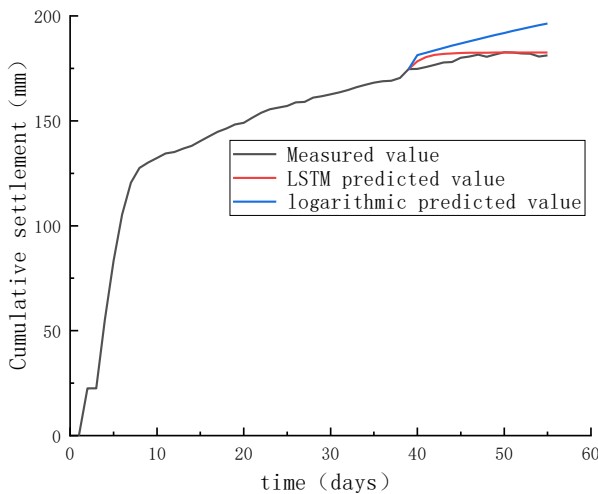

**Figure 5.** Comparison of prediction results (a).

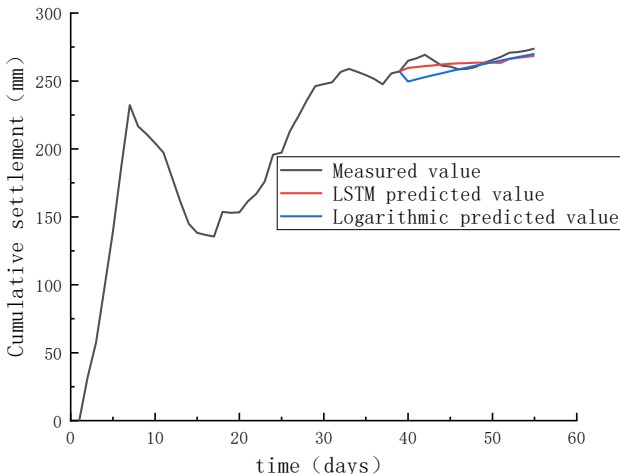

**Figure 6.** Comparison of prediction results (b).

Figure 5 shows the cumulative subsidence deformation of the vault's surrounding rock from 5 August 2020, to 28 September 2020, comprising a total of 55 data sets, out of which 39 were used for training. The data exhibits minimal fluctuations, and the surrounding rock deformation value has converged after 40 days of excavation. The model achieved convergence after about 200 training iterations, and there was no significant reduction in training loss. The optimal model obtained after model adjustment was then used to predict the remaining 16 groups of data. Similarly, Figure 6 represents the cumulative subsidence deformation of the vault's surrounding rock from 17 April 2020 to 10 June 2020, comprising a total of 55 data sets, out of which 39 were used for training. The data exhibit significant fluctuations, and the surrounding rock deformation value changes significantly from 10 to 30 days after excavation and eventually stabilizes around 50 days after excavation. The model achieved convergence after about 200 training iterations, and there was no significant reduction in training loss. The optimal model obtained after model adjustment was then used to predict the remaining 16 groups of data.

*4.2. Prediction Model*

After the model is established, the traditional curve prediction is carried out on the data of the two sections, respectively, and the prediction results are compared with those of the LSTM network model. Traditional curve prediction can select the appropriate curve type to predict the observed data and understand the trend of change in the data. Microsoft

Excel provides the ability of traditional curve prediction, and a nonlinear prediction model of tunnel vault settlement can be established in Microsoft Excel. Through comparison, it is found that the regression correlation coefficient of the logarithmic function is higher than other regression functions in predicting vault settlement. Therefore, this paper uses a logarithmic function and observed data for prediction. The comparison between the accumulated surrounding rock deformation value and the predicted value of (a) is shown in Table 2, and the comparison of the predicted results is shown in Figure 5. The comparison between the accumulated surrounding rock deformation value and the predicted value of (b) is shown in Table 3, and the comparison of the predicted results is shown in Figure 6.

**Table 2.** Measured data and predicted data (a).

| Monitoring Time /d | Measured Value /mm | Logarithmic Curve | | LSTM Model | |
|---|---|---|---|---|---|
| | | Estimate /mm | Error /mm | Estimate /mm | Error /mm |
| 40 | 174.80 | 181.31 | 6.51 | 178.42 | 3.62 |
| 41 | 175.70 | 182.48 | 6.78 | 180.46 | 4.76 |
| 42 | 176.70 | 183.62 | 6.92 | 181.39 | 4.69 |
| 43 | 177.80 | 184.74 | 6.94 | 181.87 | 4.07 |
| 44 | 178.00 | 185.83 | 7.83 | 182.15 | 4.15 |
| 45 | 180.10 | 186.89 | 6.79 | 182.30 | 2.20 |
| 46 | 180.60 | 187.93 | 7.33 | 182.40 | 1.80 |
| 47 | 181.60 | 188.95 | 7.35 | 182.46 | 0.86 |
| 48 | 180.50 | 189.95 | 9.45 | 182.50 | 2.00 |
| 49 | 181.70 | 190.93 | 9.23 | 182.53 | 0.83 |
| 50 | 182.70 | 191.89 | 9.19 | 182.55 | 0.15 |
| 51 | 182.60 | 192.83 | 10.23 | 182.57 | 0.03 |
| 52 | 182.10 | 193.75 | 11.65 | 182.58 | 0.48 |
| 53 | 182.00 | 194.65 | 12.65 | 183.59 | 1.59 |
| 54 | 180.60 | 195.54 | 14.94 | 182.60 | 2.00 |
| 55 | 181.20 | 196.41 | 15.21 | 182.60 | 1.40 |
| evaluation | RMSE | 9.7258 | | 2.6412 | |
| index | $R^2$ | 0.9312 | | 0.9791 | |

**Table 3.** Measured data and predicted data (b).

| Monitoring Time /d | Measured Value /mm | Logarithmic Curve | | LSTM Model | |
|---|---|---|---|---|---|
| | | Estimate /mm | Error /mm | Estimate /mm | Error /mm |
| 40 | 264.80 | 249.65 | 15.15 | 259.52 | 5.28 |
| 41 | 266.60 | 251.22 | 15.38 | 260.29 | 6.31 |
| 42 | 269.20 | 252.75 | 16.45 | 260.88 | 8.32 |
| 43 | 265.20 | 254.25 | 10.95 | 261.49 | 3.71 |
| 44 | 261.30 | 255.71 | 5.59 | 262.07 | 0.77 |
| 45 | 260.60 | 257.14 | 3.46 | 262.57 | 1.97 |
| 46 | 258.50 | 258.54 | 0.04 | 262.99 | 4.49 |
| 47 | 258.80 | 259.91 | 1.11 | 263.23 | 4.43 |
| 48 | 260.30 | 261.25 | 0.95 | 263.45 | 3.15 |
| 49 | 263.40 | 262.56 | 0.84 | 263.53 | 0.13 |
| 50 | 265.50 | 263.85 | 1.65 | 263.52 | 1.98 |
| 51 | 267.70 | 265.11 | 2.59 | 263.41 | 4.29 |
| 52 | 270.80 | 266.34 | 4.46 | 266.24 | 4.56 |
| 53 | 271.30 | 267.55 | 3.75 | 267.02 | 4.28 |
| 54 | 272.20 | 268.74 | 3.46 | 267.77 | 4.43 |
| 55 | 273.80 | 269.91 | 3.89 | 268.53 | 5.27 |
| evaluation | RMSE | 7.7986 | | 4.4238 | |
| index | $R^2$ | 0.7674 | | 0.8754 | |

## 5. Discussion

Based on the tunnel surrounding rock deformation data of 110 tunnel surrounding rock deformation engineering examples, this paper uses the tunnel surrounding rock deformation prediction model of the LSTM network to train and test the data. A total of 78 sets of data were used for training and 32 sets were used for testing. In order to verify the prediction effect of the established model algorithm on the measured tunnel vault settlement data, this paper selects the traditional curve prediction method and the deep learning prediction method to simulate data with two different degrees of fluctuation.

Based on the information presented in Table 2 and Figure 5, the traditional curve prediction method exhibits a maximum error of 15.21 mm, a minimum error of 6.51 mm, a root mean square error of 9.7258, and a determination coefficient of 0.9312. In contrast, the LSTM network model used in the deep learning prediction method shows a maximum error of 4.76 mm, a minimum error of 0.03 mm, a root mean square error of 2.6412, and a determination coefficient of 0.9791. The LSTM network model is significantly more effective than the traditional curve prediction method. Both the traditional and deep learning prediction methods can roughly predict the variation regularity of tunnel vault subsidence for small fluctuation data curves. However, the average error of the traditional curve prediction method is 8.85 mm, and as the forecasting period grows, the predicted results gradually tend towards a stable value. On the other hand, the LSTM network model has an average error of 2.16 mm and becomes increasingly stable and closer to the actual vault settlement as the forecasting period increases. The conventional function curve prediction method for tunnel vault subsidence is insufficient in accuracy when dealing with complex conditions. It is limited by its simplicity and a lack of parameters, making it difficult to provide precise estimates. To address this challenge, the LSTM network model is employed to predict tunnel vault subsidence based on actual training sample data. This approach is able to repair complex function relations and correct network models to enable nonlinear mapping of changes in tunnel vault subsidence. While the traditional curve prediction method can provide an approximate estimation of tunnel vault settlement data with small fluctuations, the LSTM network model in the deep learning prediction method can offer highly accurate predictions when precise tunnel vault settlement data are required.

According to the findings presented in Table 3 and Figure 6, the traditional curve prediction method exhibits a maximum error of 16.45 mm, a minimum error of 0.04 mm, a root mean square error of 7.7986, and a determination coefficient of 0.7674. In contrast, the LSTM network model used in the deep learning prediction method has a maximum error of 8.32 mm, a minimum error of 0.13 mm, a root mean square error of 4.4238, and a determination coefficient of 0.8754. The LSTM network model is significantly more accurate than the traditional curve prediction method, except for the minimum error. The traditional curve prediction method is 10.8% less accurate in predicting tunnel vault settlement data with large fluctuation ranges compared to the LSTM network model. Although the minimum error of the traditional curve prediction method is small, it fails to capture the fluctuation of the data. On the other hand, the LSTM network model captures time-series data information, detects different features during changes, and facilitates long-term learning. This model considers influencing factors as the input layer and improves prediction accuracy through backpropagation of training input and LSTM output to the hidden layer, resulting in an average prediction error of only 3.95 mm. While the traditional curve prediction method may be more accurate at specific moments when predicting tunnel vault settlement data with large fluctuations, the LSTM network model exhibits overall superior performance in comprehensive prediction.

## 6. Conclusions

During tunnel construction and operation, the deformation of the surrounding rock poses a great challenge to tunneling, and accurate prediction of changes in the tunnel structure is essential to ensure tunnel safety. For this reason, it is necessary to monitor the state of the tunnel envelope and predict its change trend. In this study, a recurrent neural network algorithm in deep learning is used to model and analyze the monitoring and measurement data of the tunnel, and an LSTM network model is established to predict the settlement of the tunnel vault. By comparison with traditional prediction methods, this study found that the LSTM model provided better prediction accuracy. The main findings are summarized as follows:

(1) The LSTM network model with deep learning is an effective method for predicting tunnel vault settlement data, offering high accuracy and stability. It outperforms traditional curve prediction methods for both data with large and small fluctuation ranges.

(2) After applying the LSTM network model to the project, the average error for small and large fluctuation data sets are 2.16 mm and 3.95 mm, respectively. The prediction model reflects the surrounding rock deformation of tunnel vault settlement well. Introducing the LSTM network model in deep learning to monitor tunnel surrounding rock deformation during construction can provide valuable insights for ensuring tunnel construction safety.

(3) Traditional prediction methods, with their limited function and parameters, are unable to accurately predict tunnel vault subsidence. The LSTM network forecasting method in deep learning overcomes this limitation through gate function control of input, memory, and output values, and backpropagation training of LSTM inputs and outputs to hidden layers, allowing it to learn and correlate time-series information for more accurate predictions of tunnel vault settlement.

(4) Traditional methods for monitoring surrounding rock deformation in tunnels are inefficient, imprecise, and require significant manpower and resources. The future of tunnel engineering demands accurate and fast forecasting of surrounding rock deformation, and the application of artificial intelligence in tunnel engineering, particularly deep learning algorithms, can play a significant role in ensuring tunnel safety. The prediction results obtained through deep learning algorithms can provide feedback for tunnel construction, optimize construction parameters, and facilitate proactive protective measures.

**Author Contributions:** Writing—original draft, Y.H.; Writing—review & editing, Q.C. All authors have read and agreed to the published version of the manuscript.

**Funding:** This work is supported by the National Natural Science Foundation of China (No. 52078211) and the Natural Science Foundation of Hunan Province (2020JJ4021).

**Institutional Review Board Statement:** Not applicable.

**Informed Consent Statement:** Informed consent was obtained from all subjects involved in the study.

**Data Availability Statement:** Not applicable.

**Conflicts of Interest:** The authors declare no conflict of interest.

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
