# Peer review of "Construction and Application of LSTM-Based Prediction Model for Tunnel Surrounding Rock Deformation"

_sustainability, doi:10.3390/su15086877_

Round 1

Reviewer 1 Report

An interesting and well presented study.

The research question is well articulated. The methodology is appropriate to the subject of the research and the development of the LDSM model is explained in the necessary detail. The methodology and discussion are supported by a adequate review of the relevant literature.

The Conclusions are consistent with with the evidence and arguments presented. 

The model that is developed and the research findings will be relevant to other academics in the field and inductry practitioners.

Author Response

We deeply appreciate the time and effort you’ve spent in reviewing our manuscript (Manuscript Number: sustainability-2293747). Thank you very much for constructive suggestions and comments, which greatly improve our manuscript.
Thank you for your approval of my manuscript, I have made modest changes to the English language in the manuscript, we mark all the changes in red in the revised manuscript. Please see the attachment.

Reviewer 2 Report

The authors have talked in their abstract about LSTM network model, readers may have no clue what LSTM is and what stands for. They should have explained it first. 

The first line in the introduction it is written " In recent years, China has promote its .... etc", which should be " In recent years, China has promoted its .... etc" 

The authors in the introduction have stated that " In 1985 (the researcher name) ... In 1984 (the researcher name)  ... In 2006 (the researcher name) and then suddenly turned it to In China Professor Zhang Qing [2]. I guess the way they refer to the researchers must be consistent. 

The main problem of LSTM is that the long-term information has to sequentially travel through all cells before getting to the present processing cell. Which means it can be easily corrupted by being multiplied many time when it gets multiplied by numbers less than zero, which is the main cause of  vanishing gradients. Also, RNN and LSTM are difficult to train because they require memory-bandwidth-bound computation, have the authors overcome this issue?

Figure 5. Cumulative value of tunnel crown settlement is not clear.

At which times the training loss was significant ? 

The paragraph from Line 244 to 250 is not clear and needs to be revised.

Author Response

(The authors gave the same response as above.)

Reviewer 3 Report

Dear Authors, I have read this text and below are my comments:

Lack of a "discussion"-formal error of the text.

The authors themselves cite many people and their studies in the introduction and the references to their works are missing, meanwhile the amount of literature cited is only 19 items.

The cited formulas must have all variables explained.

There are minor typos in the text, e.g. no space, lines 139

In summary, this text needs to be improved, written in a more accessible way for the reader.

Better document your data in the context of existing knowledge (which can be done in the Discussion section).

Author Response

(The authors gave the same response as above.)

Reviewer 4 Report

Point 1: The introduction section needs a systematic review of related studies and needs to provide a clear connection with those cited papers in the first paragraph. The reviewer suggests providing a tabulated literature review comparing different methods and (or) prediction models for tunneling-related rock deformation with their pros and cons.

Point 2: From lines 34 through 39, the reviewer suggests citing related and recent papers.

Point 3: Kindly follow the correct format for citing authors.

Point 4: Kindly define any acronyms and (or) abbreviations during their first appearance in the abstract section and main body.

Point 5: From lines 68 through 73, kindly explicitly emphasize the research gap.

Point 6: From lines 74 through 76, the reviewer suggests citing related and recent papers.

Point 7: Section 2 is challenging to understand as the figures provided must be explained better. The figures provided need to be elaborated explicitly in each sub-section. The symbols and (or) nomenclatures need to be also improved.

Point 8: The reviewer strongly suggests providing more details about the tunnel under consideration, specifically the monitoring program. Some monitoring information appears out of nowhere. Kindly provide one section or subsection for the tunneling details.

Point 9: How did the authors decide on partitioning the training and testing data?

Point 10: In line 128, what other models were the authors referring to? The authors only did curve fitting rather than using tunneling-related ground deformation prediction models (i.e., logistic model, Gompertz model, etc.).

Point 11: Tables 2 and 3 are unnecessary since the values are plotted in Figures 6 and 7. Likewise, Figure 5 seems redundant with Figures 6 and 7.

Point 12: The presentation of results needed to be more profound, and the authors needed to come up with other means of presenting their results.

Point 13: The authors also need to provide a more elaborate discussion regarding the comparison and limitations of the proposed prediction model.

Point 14: Kindly follow the correct format of listing references.

Author Response

(The authors gave the same response as above.)

Round 2

Reviewer 2 Report

The comments have been addressed. 

Author Response

Response to Reviewer and Editor Comments

Dear editors and reviewer#2,

We deeply appreciate the time and effort you’ve spent in reviewing our manuscript (Manuscript Number: sustainability-2293747). Thank you very much for constructive suggestions and comments, which greatly improve our manuscript.

With best wishs,

Sincerely

Reviewer 3 Report

Dear Authors, thank you for your work on the manuscript. In my opinion, he is much better now. I have no more objections.

Author Response

Response to Reviewer and Editor Comments

Dear editors and reviewer#3,

We deeply appreciate the time and effort you’ve spent in reviewing our manuscript (Manuscript Number: sustainability-2293747). Thank you very much for constructive suggestions and comments, which greatly improve our manuscript.

With best wishs,

Sincerely

Reviewer 4 Report

The reviewer appreciates the effort of the authors in promptly revising the paper. The paper has partly improved after the first revision. However, some of the critical comments and suggestions remain. Additional comments and suggestions are listed below.

Point 1: The reviewer strongly recommends removing Tables 2 and 3 and including the overall statistical results in Figures 6 and 7 (i.e., RMSE, R2, MaxError, and MinError). Likewise, Figure 5 needs to be removed as it is basically replotted in Figures 6 and 7.

Point 2: The reviewer strongly suggests that the authors use other means to present the results. In other words, Figures 6 and 7 are insufficient to support the novelty of the employed LSTM-based prediction model. Why did the authors start the modeling only on day 40?

Point 3: The abstract should include quantitative results and (or) findings rather than a broad summary.

Point 4: The authors needed to follow the correct citation format. Kindly check other papers published in Sustainability or other MDPI journals and study how to cite other authors' work properly.

Point 5: In the introduction section, kindly briefly overview the paper's organization.

Point 6: Ensure that all figures (i.e., fonts and (or) labels) are all legible. Kindly increase the font size to almost equal the font size of the text in the manuscript.

Point 7: The authors needed to follow the correct format of listing references. Kindly check other papers published in Sustainability or other MDPI journals and study how to list references properly.

Author Response

Response to Reviewer and Editor Comments

Dear editors and reviewer#4,

We deeply appreciate the time and effort you’ve spent in reviewing our manuscript (Manuscript Number: sustainability-2293747). Thank you very much for constructive suggestions and comments, which greatly improve our manuscript.

Reviewers’ comments are listed here:

Review comment 1: 

The reviewer strongly recommends removing Tables 2 and 3 and including the overall statistical results in Figures 6 and 7 (i.e., RMSE, R2, MaxError, and MinError). Likewise, Figure 5 needs to be removed as it is basically replotted in Figures 6 and 7.

  • Response:

Thank you for the comment. Considering your suggestion, Figure 5 has been removed. Tables 2 and 3 contain specific data with associated evaluation indices that provide a more comprehensive overview of the changes in tunnel deformation. Provide specific values that can be used by other scholars for model calculation comparison.

Review comment 2: 

The reviewer strongly suggests that the authors use other means to present the results. In other words, Figures 6 and 7 are insufficient to support the novelty of the employed LSTM-based prediction model. Why did the authors start the modeling only on day 40?

  • Response:

Thank you for your comments. In this study, there are 55 data sets for each model group, where the first 39 days are the training set data and after 40 days are the test set data. The training set was chosen to represent 70% of the entire sample, while the remaining 30% was designated as the test set [1, 2]. This method is commonly used by scholars, and therefore it was adopted in this study.

[1] LI L B, G X, GAN X L, et al. Prediction of maximum ground settlement induced by shield tunneling based on recurrent neural network [J]. Chinese Journal of Civil Engineering, 2020, 53(Suppl. 1): 13-19.

[2] LIU Z B, Li L, Fang X L, et al. Hard-rock tunnel lithology prediction with TBM construction big data using a global-attention-mechanism based LSTM network[J]. Automation in Construction,2021,125.

Review comment 3: 

The abstract should include quantitative results and (or) findings rather than a broad summary.

  • Response:

Thank you for your comment. The abstract has been rewritten and improved to take into account your suggestions. Please see Lines 10-28.

That is,

Tunnel surrounding rock deformation is a significant issue in tunnel construction and maintenance, which has garnered attention from both domestic and international scholars. Traditional methods of predicting tunnel surrounding rock deformation involve fitting monitoring and measuring data, which is a laborious and resource-intensive process with low accuracy when predicting data with significant fluctuations. Deep learning approach can improve monitoring efficiency and accuracy while reducing labor costs. In this study,taking an actual tunnel project as an example, the Long Short-Term Memory (LSTM) network model was constructed based on the recurrent neural network algorithm in deep learning to model and analyze the tunnel monitoring and measurement data, and the model was used to analyze and predict the vault settlement of the tunnel. LSTM is a type of artificial neural network architecture that is commonly used in deep learning applications for sequence prediction tasks, such as natural language processing, speech recognition, and time-series forecasting. In predicting the data with smaller fluctuations, the maximum error is 4.76 mm, the minimum error is 0.03 mm, the root mean square error is 2.6412, and the coefficient of determination is 0.9791. In predicting the data with larger fluctuations, the maximum error is 8.32 mm, the minimum error is 0.13 mm, the root mean square error is 4.4238, and the coefficient of determination is 0.8754. The average error of the LSTM network model is 2.16 mm. With the growth of the prediction period, the prediction results become more and more stable and closer to the actual vault settlement.  which provides a reliable reference for introducing the LSTM prediction method in deep learning to tunnel construction and promoting tunnel construction safety.

Review comment 4:

The authors needed to follow the correct citation format. Kindly check other papers published in Sustainability or other MDPI journals and study how to cite other authors' work properly.

  • Response:

Thank you for your comment. Taking your suggestion into account, we have referred to the paper in Sustainability and made changes to the works cited by other authorsPlease see Lines 49-69.

That is,

In China, Zhang et al. [7] took the lead in introducing artificial intelligence theory into rock mechanics and geotechnical engineering, and predicted the mechanical properties of rocks. Song et al. [8] proposed the observation and research method of underground rock strata dynamics, the prediction and control mechanical model, and the roof prediction and control technology. Feng et al. [9] put forward the concept of intelligent rock mechanics, found out the internal relations contained in engineering instance data, used these relations to make a reasonable judgment on tunnel stability, and established an intelligent system for rockburst risk estimation. Chen et al. [10] pointed out that modern information technology such as big data should be used to strengthen the construction of tunnel database and establish intelligent data monitoring. Chen et al. [11] summarized the main research progress of stability analysis, slope intelligent monitoring and slope intelligent prediction of artificial intelligence technology in geotechnical engineering, and put forward prospects for solving slope stability problems with artificial intelligence technology. Wang et al. [12] carried out the physical and mechanical test research of soft rock, and combined with finite element numerical simulation, analyzed and predicted the deformation of tunnel surrounding rock. Jiang [13] used artificial intelligence technology to predict rock mechanical behavior, and proposed particle swarm optimization support vector machine model for rock mechanical behavior prediction. Hu et al. [14]

Review comment 5: 

In the introduction section, kindly briefly overview the paper's organization.

  • Response:

Thank you for your comment. We have taken your suggestion into consideration and we present the organization of the paper in the abstract and introduction. Please see Lines 15-19, 81-84.

That is,

In this study,taking an actual tunnel project as an example, the Long Short-Term Memory (LSTM) network model was constructed based on the recurrent neural network algorithm in deep learning to model and analyze the tunnel monitoring and measurement data, and the model was used to analyze and predict the vault settlement of the tunnel.

To address this issue, this study focuses on a particular tunnel construction project and collects monitoring data during construction. Using LSTM networks, a targeted prediction model is constructed and validated through comparisons with existing models.

Review comment 6: 

Ensure that all figures (i.e., fonts and (or) labels) are all legible. Kindly increase the font size to almost equal the font size of the text in the manuscript.

  • Response:

Thank you for your careful review. The font size of the text in the manuscript has been checked

Review comment 7: 

The authors needed to follow the correct format of listing references. Kindly check other papers published in Sustainability or other MDPI journals and study how to list references properly.

  • Response:

Thank you for your comments. Taking your suggestions into account, we have modified the format of the references in this paper by referring to the format of the paper references in Sustainability.

Thank you again to the editor and reviewers for your suggestions.We hope that these revisions meet with your approval, and we look forward to hearing back from you. Thank you again for your thoughtful review.

With best wishs,

Sincerely

Round 3

Reviewer 4 Report

The reviewer appreciates the authors' effort in revising the paper. Some of the comments and suggestions were addressed and accepted by the reviewer. Minor revisions are listed below.

Point 1: In the abstract, kindly use only two decimal places for the quantitative results.

Point 2: In the introduction section, kindly briefly overview the paper's organization. Check other published articles in Sustainability on how to provide a paper's structure preview.

Point 3: Ensure that all figures (i.e., fonts and (or) labels) are all legible. Kindly increase the font size to almost equal the font size of the text in the manuscript. Specifically, the labels and legends from Figure 3 onwards are not legible.

Point 4: The authors needed to follow the correct format of listing references. Kindly check other papers published in Sustainability or other MDPI journals and study how to list references properly.

Point 5: Kindly proofread the paper carefully.

Author Response

Response to Reviewer and Editor Comments

Dear editors and reviewer#4,

We deeply appreciate the time and effort you’ve spent in reviewing our manuscript (Manuscript Number: sustainability-2293747). Thank you very much for constructive suggestions and comments, which greatly improve our manuscript.

Please see the attachment for details.
